# The Expression of Carbonic Anhydrases II, IX and XII in Brain Tumors

**DOI:** 10.3390/cancers12071723

**Published:** 2020-06-29

**Authors:** Joonas Haapasalo, Kristiina Nordfors, Hannu Haapasalo, Seppo Parkkila

**Affiliations:** 1The Hospital for Sick Children, Arthur and Sonia Labatt Brain Tumor Research Center, Toronto, ON M5G 0A4, Canada; Kristiina.nordfors@sickkids.ca; 2Unit of Neurosurgery, Tampere University Hospital, FI-33520 Tampere, Finland; 3The Hospital for Sick Children, Division of Hematology and Oncology, Toronto, ON M5G 0A4, Canada; 4Faculty of Medicine and Health Technology, Tampere University, FI-33520 Tampere, Finland; hannu.haapasalo@fimlab.fi (H.H.); seppo.parkkila@tuni.fi (S.P.); 5Fimlab Laboratories Ltd., FI-33520 Tampere, Finland

**Keywords:** carbonic anhydrase, brain tumor, glioma, astrocytoma, oligodendroglioma, medulloblastoma, prognosticator, carbonic anhydrase inhibitor

## Abstract

Carbonic anhydrases (CAs) are zinc-containing metalloenzymes that participate in the regulation of pH homeostasis in addition to many other important physiological functions. Importantly, CAs have been associated with neoplastic processes and cancer. Brain tumors represent a heterogeneous group of diseases with a frequently dismal prognosis, and new treatment options are urgently needed. In this review article, we summarize the previously published literature about CAs in brain tumors, especially on CA II and hypoxia-inducible CA IX and CA XII. We review here their role in tumorigenesis and potential value in predicting prognosis of brain tumors, including astrocytomas, oligodendrogliomas, ependymomas, medulloblastomas, meningiomas, and craniopharyngiomas. We also introduce both already completed and ongoing studies focusing on CA inhibition as a potential anti-cancer strategy.

## 1. Introduction

Carbonic anhydrases (CAs) are ubiquitous metalloenzymes that are present in both prokaryotes and eukaryotes from archaea and bacteria to plants and animals [1]. The role of CAs in the regulation of pH homeostasis has been known for almost a century [2]. CAs catalyze the key chemical reaction for pH regulation: CO_2_ + H_2_O <=> HCO_3_^-^+ H^+^. The CA gene family consists of eight unrelated enzyme families (alfa, beta, gamma, delta, zeta, eta, theta, and iota). Mammals have only CAs belonging to the alfa class, and their functions are essential in many physiological processes, e.g., in gluconeogenesis, lipogenesis, and ureagenesis. Importantly, CAs participate in various physiological roles, such as regulation of intra-and extracellular pH and ion transport, as well as water and electrolyte balance.

The human alfa-CA family involves 15 members which can be divided according to their cellular location (intracellular or extracellular) or enzymatic activity [3]. Five active family members are located in the cytosol (CA I–III, VII, and XIII), four are membrane-associated (CA IV, IX, XII, and XIV), two are mitochondrial (CA VA and VB), and one is a secretory form (CA VI). In addition, there are three acatalytic forms, which are called CA-related proteins (CARPs) [4,5,6]. 

Many CAs have been associated with neoplastic growth. Especially the hypoxia-inducible CA IX and CA XII as well as the widely expressed CA II have been reported in certain tumors [6,7,8]. It has been suggested that CAs contribute to stabilization of intracellular pH of tumor cells like in other cells, and consequently also contribute to extracellular acidification that is attributable to the high metabolic rate of tumor cells, producing both excessive carbon dioxide and lactic acid to be removed from the cells (Figure 1). Extracellular acidification has been functionally linked to the malignant behavior of cancer cells [9]. 

Brain tumors form a heterogeneous group of diseases that include both benign and malignant tumors. In this review, we describe some of the recent knowledge on CAs in brain neoplasms, and focus on isozymes II, IX, and XII, which are currently considered potential diagnostic biomarkers and therapeutic targets.

## 2. CAs and Their Physiology in Normal CNS

### 2.1. Physiological Functions of CAs in CNS

The finding that CA activity plays a role in human brain physiology was described by Asby in 1943 [11]. Since then there have been numerous publications about CA activity and the roles of CA enzymes in the central nervous system (CNS). The functions of CAs have been studied in several animal models, and it has been shown that CAs play multiple roles, e.g., in formation of cerebrospinal fluid (CSF) [12], extracellular fluid and ion homeostasis [13], epileptic seizure activity [14], respiratory response to carbon dioxide [14], generation of bicarbonate for biosynthetic processes [15], and neuronal signal transduction [16,17]. CA IX may play a role in the regulation of behavior and maintenance of tissue integrity in the brain since CA IX-deficient (*Car9*^−/−^) mice were reported to have mild behavioral changes and morphological disruption of brain histology [18].

### 2.2. CA Expression in Normal Brain

#### 2.2.1. CA Expression Based on Animal Models

The choroid plexus is one of the sites in CNS where CAs are abundantly expressed. Early physiological studies using a pan-CA inhibitor, acetazolamide, showed that CA inhibition reduced cerebrospinal fluid (CSF) production in the choroid plexus [19,20]. These findings introduced acetazolamide as a potential drug for the treatment of brain edema. Later studies have demonstrated CA II as the key enzyme that is abundantly expressed in the choroid plexus [21]. 

Early histochemical studies in mice showed that the highest CA activity was present in brain areas that are rich in myelinated fibers and glial cells [22]. Immunohistochemical staining has confirmed that CA II is mainly expressed in the oligodendrocytes [21,23,24,25,26], whereas the expression of CA II in astrocytes has some variability and these glial cells may contain the enzyme at relatively low levels [23,27,28]. When studied by in situ hybridization, CA II mRNA expression was only found in the oligodendrocytes among mouse primary cultured glial cells [29]. In addition, CA II has been found in myelin sheaths, which is based on the high expression in oligodendrocytes [21,23]. Neurons may generally lack CA II, but at least some trigeminal ganglion neurons express it in the rat brain [30]. Reactive microglial cells were heavily immunostained for CA II in the spinal cord injury rat model [31]. The expression of different CA isozymes in the CNS is presented in Table 1. 

#### 2.2.2. CA Expression in Human Brain Tissue

In the normal human brain, CA II is clearly present in the oligodendrocytes and choroid plexus epithelium, but the expression was reported to decline during aging [37]. CA II was also seen in a subset of neurons mostly with GABAergic phenotype, in a few astrocytes, and transiently during brain development in the endothelial cells of microvessels. CA IX and CA XII were first studied in the human brain at a transcriptional level [44]; CA IX mRNA was nearly absent in the brain (e.g., cerebrum, cerebellum, ventricle, pons, and pituitary gland), but CA XII mRNA produced strong hybridization signals in the putamen and caudate nucleus, whereas signals from other brain regions were faint. Ivanov and colleagues also studied the expression of both enzymes by immunohistochemistry and found that CA IX was limited to the ventricular lining cells and the choroid plexus, whereas CA XII immunoreactivity was restricted to the posterior lobe of the pituitary glands, a remnant of Rathke’s pouch, the choroid plexus, and limited numbers of ganglion cells in the cortex [44].

## 3. CAs in Brain Tumors

### 3.1. Screening for CA II, CA IX and CA XII in Different Brain Tumors

The first report on CAs in brain tumors involved immunohistochemical staining of CA II in a series of archived tumor specimens [45]. CA II was found to be present in several tumors including astrocytomas, oligodendrogliomas, and medulloblastomas, even though these early studies involved a very limited number of specimens. In addition, some acoustic neurinomas, a plexiform neurofibroma, a choroid plexus papilloma, an ependymoblastoma and a subependymoma were positive for CA II. 

For CA IX and CA XII, Ivanov and coworkers screened various cell lines and tumors with different genetic backgrounds including brain tumors [44]. In vitro glioma models revealed high-to-moderate levels of CA IX and XII mRNAs. CA IX expression was further studied by immunohistochemistry in a small series of gliomas, showing that low-grade gliomas and oligodendrogliomas were negative for CA IX, whereas grade III-IV gliomas were all CA IX-positive. In addition, all hemangioblastomas (three tumors), meningiomas (five tumors), and two out of three choroid plexus tumors were positive for CA IX.

Proescholdt and colleagues conducted another study to investigate the levels of hypoxia-inducible CA IX and CA XII in brain tumors with different histology and malignancy grade [46]. They analyzed tissue specimens from both brain tumors and normal brain by immunohistochemistry, Western blot, and in situ hybridization. They examined 112 tumor samples (astrocytomas, meningiomas, metastases, primitive neuroectodermal tumors (PNETs), and hemangioblastomas). In the glioma specimens, the strongest staining for CA IX was detected in glioblastomas (97% tumors were positive). In addition, the expression in tumors was found to be upregulated in concordance with increasing malignancy grade. Similarly, stronger CA XII staining was associated with more aggressive tumors. The results also demonstrated that the expression of both isozymes was linked with necrosis. The most widespread CA IX and XII staining of all tumors was detected in hemangioblastoma samples affected by non- functioning von Hippel-Lindau protein. The cerebral metastases from tumors outside the CNS showed strong staining for CA IX and XII. 

In Figure 2, Figure 3, Figure 4, Figure 5 and Figure 6, we present examples of immunohistochemical staining for CA II, CA IX, and CA XII in grade II-III astrocytomas, grade IV glioblastomas, medulloblastomas, meningiomas, and normal brain. The images clearly demonstrate that CA IX immunostaining is most predominant, although it shows a lot of variation both within and between tumor specimens. CA II and CA XII typically show faint or moderate reactions in different tumor categories. 

### 3.2. Expression of CA II, CA IX and CA XII in Pediatric Brain Tumors

There are few articles of CA IX expression in pediatric brain tumors. Ivanov and coworkers screened brain tumors by immunohistochemistry and found that six of the seven central/peripheral PNETs expressed CA IX, and all of the six studied ependymomas were positive for the isozyme [44]. Preusser and colleagues assessed CA IX in intracranial ependymomas: 84 out of 100 tumors expressed CA IX, and it was associated with bizarre angiogenesis and necrosis [47]. CA IX expression was not associated with patient survival, although high hypoxic score (combined CA IX, HIF-1, and VEGF) was associated with poor prognosis. 

Our team investigated the expression of CA II, CA IX, and CA XII in a series of WHO grade IV medulloblastoma/PNET specimens (*n* = 39) [48]. Immunohistochemical results showed that endothelial CA II, cytoplasmic CA II, CA IX, and CA XII were expressed in 49%, 73%, 23%, and 11% of the tumors, respectively. Similar to gliomas [49], CA II was expressed in the neovessel endothelium, but also to some extent in the tumor cell cytoplasm [48]. Positive signal for CA IX was located in tumor cells near necrosis, and importantly, the expression predicted a poor outcome in both univariate and multivariate analyses. These results prompted us to propose that the use of CA inhibitors could represent a promising treatment option for certain CA-positive medulloblastomas. 

### 3.3. CA II in Diffuse Astrocytic and Oligodendroglial Tumors

Since cytosolic CA II is expressed mainly in the oligodendrocytes and its expression is induced in the endothelium of neovessels in several cancers [50], a study was designed to assess the endothelial expression of CA II in a series of 255 diffuse astrocytic and 71 oligodendroglial tumor specimens [49]. Endothelial CA II expression was either absent or weak in grade II diffuse gliomas, while grade 3 mixed oligoastrocytoma and glioblastoma (grade IV) specimens were the most positively stained tumor types. Survival analysis indicated that endothelial CA II staining was associated with poor prognosis in patients with diffuse astrocytic tumors. The presence of CA II in the tumor endothelium suggests that it may play an important role in the metabolic processes of tumor cells.

### 3.4. CA IX in Astrocytomas

The first large study to evaluate CAs as prognosticators of brain tumors was published in 2006 [51]. 362 diffuse astrocytic tumors (grades II-IV) were studied by immunohistochemistry and the protein level immunostaining results of six tumor specimens (two tumors from each grade 2–4) were verified by means of mRNA analysis. Cellular CA IX immunopositivity was observed in 78% of tumors (65% grade II diffuse astrocytomas, 73% of grade III anaplastic astrocytomas, and 82% of grade IV glioblastomas). Similarly, CA IX expression correlated to the increasing WHO grade, presence of necrosis, and the staining was seen in the infiltrative neoplastic tumor cells. Most importantly, CA IX expression predicted a poor outcome of patients with diffuse astrocytic tumors in both univariate and multivariate analyses, highlighting the importance of CA IX in gliomagenesis. More specifically, CA IX predicted a poor prognosis in both glioblastoma (grade IV) and grade II diffuse astrocytoma.

The prognostic significance of CA IX in diffuse astrocytic tumors has been later confirmed by others. First, Korkolopoulou and coworkers studied the expression of hypoxia-related tissue factors in astrocytic gliomas (*n* = 84), of which 72.6% were immunopositive [52]. In multivariate analysis, CA IX with grade and age were the only parameters affecting patient survival. CA IX expression was also the only significant parameter for the survival of patients with grades II/III. Furthermore, CA IX immunoreactivity had perinecrotic distribution and increased in parallel with the extent of necrosis and histologic grade. 

Second, Sathornsumetee and coworkers conducted a retrospective analysis of 68 patients with recurrent malignant gliomas (35 patients with glioblastoma and 33 patients with WHO grade III gliomas), who were previously treated in a phase II trial of bevacizumab and irinotecan (https://clinicaltrials.gov/ct2/show/NCT00268359) [53]. The study population consisted of recurrent malignant glioma patients with measurable recurrent or residual primary disease, already treated with chemotherapy with or without radiotherapy. Thus, the patients had tumors affected by cancer treatment and these recurrent tumors could be considered a model to study clonal selection. Again, positive CA IX immunostaining was associated with poor survival outcome in both univariate and multivariate analyses.

Third, Yoo, and co-workers investigated 78 WHO grade II, III, and IV astrocytic gliomas by immunohistochemistry, and found that CA IX expression was associated with increasing WHO grade and poor survival in grade II-IV tumors, as well as with poor survival in WHO grade II tumors when these tumors were evaluated separately [54]. 

Fourth, Flynn and colleagues evaluated hypoxia-regulated proteins, including CA IX, and survival with adult glioblastomas, but did not find any statistically significant association [55]. Notably, they used a commercial antibody against CA IX instead of the well characterized M75 antibody which has been used by numerous investigators and shown as highly specific against CA IX [51,52]. In their analysis, the immunohistochemical score for CA IX remained at a relatively low level (a little over 2 on scale 0–4). This could be due to low reactivity of the antibody or specific scoring system (counting only the percentage of positive cells without taking into account the staining intensity). Even then the higher CA IX signal showed a slight, although non-significant, correlation with poorer survival. 

Fifth, Proecholdt et al. showed that CA IX was an independent prognostic factor for dismal outcomes in patients with glioblastoma [56]. 

Sixth, Erpolat and colleagues demonstrated that CA IX, evaluated as either part of hypoxic profile molecules or as a single factor, correlated to shorter survival of high-grade glioma patients (*n* = 172, grade III-IV astrocytomas) [57]. They further suggested that a combination of hypoxic markers, involving CA IX, HIF-1α and osteopontin, is more robust than a single marker for predicting survival in high-grade glioma.

Lastly, Cetin and coworkers verified the finding of prognostic value of CA IX in glioblastomas and showed that the median overall survival was longer in patients with low levels of CA IX expression (18 months) compared to patients with CA IX overexpression (9 months) [58]. In multivariate analysis, significant prognosticators included CA IX overexpression, Karnofsky performance scale, incomplete temozolomide treatment, and gross-total resection. There was a trend towards a longer median progression-free survival in patients with low CA IX expression (8 months vs. 3.5 months), but this difference did not reach statistical significance (*p* = 0.054). 

As a conclusion from the several articles presented above, CA IX expression is a common phenomenon in malignant astrocytomas, and it can be used as a prognosticator in grade II-IV diffuse astrocytic tumors, and separately in glioblastomas. This was also the conclusion in one meta-analysis regarding CA IX and all tumor types [59].

### 3.5. CA IX in Oligodendroglial Tumors

Another important group of gliomas is oligodendroglial tumors. Birner and coworkers analyzed 44 primary and 16 recurrent oligodendroglial neoplasms with 1p deletion or imbalance status for hypoxia-inducible factor 1α (HIF1α), CA IX, and vascular endothelial growth factor (VEGF), and based on the expression of all molecules a hypoxia score was established [60]. Although the hypoxia score did not correlate significantly to patient survival in univariate analysis, CA IX expression was associated with poor survival in the subgroup of patients who received adjuvant therapy (but not with the whole material). Furthermore, oligodendroglial tumors have been shown to express CA IX in 80% of the neoplasms and to be an independent prognostic indicator in a material of 86 grade II-III tumors [61]. Abraham and coworkers also evaluated the prognostic value of CA IX in 50 patients with oligodendrogliomas and 32 oligoastrocytomas and found an adverse prognostic correlation in oligoastrocytomas also in multivariate analysis, but not in oligodendrogliomas [62]. These three original articles show that CA IX plays a central role in oligodendrogliomas and could perhaps be used as a prognosticator. The challenge for the latter is that in WHO 2016 classification, the diagnosis of oligoastrocytoma is discouraged (The 2016 World Health Organization Classification of Tumors of the Central Nervous System) [63].

### 3.6. CAs in Other Brain Tumors and Metastases

After Proecholdt et al. [46], both CA IX and CA II have been studied in various other brain tumors. The prognostic correlations of CA XII have been studied by us in diffuse astrocytic tumors [64]. Of 370 neoplasms, 98% were immunopositive to CA XII and the expression correlated with worse patient prognosis in both univariate and multivariate survival analyses. Furthermore, it was shown that CA XII present in diffuse astrocytic tumors is mainly encoded by a shorter mRNA variant being possibly linked to the aggressive behavior of the tumor. To our knowledge, there are no other studies on the prognostic significance of CA XII in brain tumors.

CAs have also been studied in meningiomas, which are mostly benign tumors but can occasionally behave malignantly. Immunohistochemical staining indicated that 50% of all meningiomas contain CA IX-positive hypoxic regions, and the expression is significantly associated with higher-grade histology [65]. Both CA II and IX expression levels have been immunohistochemically evaluated in a large series of meningiomas (*n* = 443 primary and 67 recurrent tumors) [66]. 14.8% of the tumors stained positively for CA II in tumor endothelium, whereas 11.6% of tumor cells were positive for CA IX. Endothelial CA II expression correlated with increasing histological grade and tumor proliferation rates, suggesting that CA II expression is associated with malignant progression of meningiomas, whereas there was neither association of CA IX to clinicopathological features nor patient survival. However, Jensen and coworkers found correlations; intracranial meningiomas with high CA IX expression were often of higher malignancy grade, although the expression did not predict survival [65]. As a conclusion, there is at least some evidence that CA II and IX expression levels correlate with the progression of meningiomas towards a more malignant phenotype.

Craniopharyngiomas represent still another tumor type where CA IX is significantly upregulated. CA IX staining was associated with increased cyst size when studied in 20 craniopharyngiomas [67]. Since cystic growth can cause pressure of the adjacent brain and is often the cause of morbidity, it was suggested that CA IX inhibitors may serve as potential adjuvant drugs for the treatment of patients with cystic craniopharyngiomas.

There are very few studies available on the expression of CAs in brain metastases. CA IX expression shows high correlation between primary and secondary cancers at least in the case of non-small cell lung carcinoma and matched brain metastases [68]. Recently, Prayson, and colleagues investigated CA IX expression in clear cell meningiomas which are difficult to distinguish from metastatic clear cell renal cell carcinomas [69]. The study material included 18 patients with clear cell meningiomas and 26 cases of clear cell renal cell carcinomas. 38.9% of clear cell meningiomas and 93.8% clear cell renal cell carcinomas were CA IX-positive. The authors proposed that CA IX immunostaining could be of potential value with other markers, such as CD10 and renal cell carcinoma antigen, for differential diagnostics of clear cell meningiomas from metastatic clear cell renal cell carcinoma. 

Recently Zheng and coworkers analyzed a series of 34 metastatic renal cell carcinomas, 30 hemangioblastomas, 11 microcystic meningiomas, and 3 clear cell meningiomas for the expression of potential biomarkers. They showed that an immunohistochemical panel involving CA IX, PAX8, SST2Ra, and inhibin could be useful for histopathological diagnostics of these tumor entities [70]. 

## 4. Use of CA Inhibitors in Brain Tumors

### 4.1. Pre-Clinical Studies

To our knowledge, there are few published reports on the use of CA inhibitors in pre-clinical models of brain tumors. The CA inhibitors with most data include the classical drug, acetazolamide, anticonvulsants (topiramate and zonisamide), and a CA IX/CA XII inhibitor, SLC-0111. 

The FDA-approved indications for acetazolamide are altitude sickness, glaucoma, congestive heart failure, idiopathic intracranial hypertension, periodic paralysis, and epilepsy, and several off-label indications have been described [71]. Acute clinical doses of acetazolamide reduce body fluid compartments, can be used as diuretics, and lead to a metabolic acidosis [72]. Common side effects include e.g., fatigue, nausea, vomiting, and paresthesia. The re-positioning of the drug to cancer therapy would be an attractive strategy, especially because there is preclinical evidence for the combination of acetazolamide with anti-tumor drugs [73,74,75,76,77,78]. 

Amiri and colleagues recently studied the effect of acetazolamide alone or in combination with temozolomide (TMZ) in 2D and 3D cell culture models of glioblastoma cells (U251N) and human brain tumor stem cells [79]. Cell death was significantly increased with combined treatment in both models. The effect of acetazolamide was further induced by incorporation into nano-carriers. Notably, the effect of acetazolamide may not be limited to inhibition of CA activity, but it may also decrease CA IX mRNA and protein expression in glioblastoma cells [74].

In another experimental study in mice, Albatany and coworkers investigated the effect of intraperitoneal injection of five drugs (acetazolamide, quercetin, cariporide, dicholoroacetate, and pantoprazole) for intracellular pH regulation in implanted U87 glioblastoma cells [80]. Combination drug injection resulted in a decrease of intracellular pH (estimated as 0.4 pH drop). Injection of the drug combination with glucose produced even greater reduction in intracellular pH (estimated as 0.72 pH drop).

Recently, Hannen and colleagues reported that CA II is upregulated in TMZ resistant stem-like cells compared to control cells [81]. The same phenomenon was observed in clinical specimens of first manifest and recurrent tumors. Importantly, treatment of the cells with acetazolamide sensitized them to TMZ-induced cell death. 

Topiramate is a sulfamate-substituted monosaccharide which was originally found to exhibit potent anticonvulsant activity similar to phenytoin [82]. The first report described topiramate as a weak CA inhibitor (micromolar against erythrocyte CAs), but later Supuran and coworkers demonstrated that topiramate is, in fact, a very potent CA inhibitor with the Ki value 5 nM against human CA II [83]. Topiramate efficiently inhibits the major CA isozymes present in the brain, causing CO_2_ retention which is considered important for the anticonvulsant effect [84]. Other mechanisms may also be involved, such as blockade of Na^+^ channels and AMPA/kainate receptors, as well as enhancement of GABAergic transmission. The effect of topiramate on intratumoral pH regulation has been investigated in a study where U87 glioblastoma cells were injected into the immunocompromised mouse brain and single dose of topiramate was given prior to the measurement of tissue pH [85]. Intraperitoneal topiramate administration induced significant intracellular acidification in the implanted brain tumors, whereas contralateral brain tissue showed no pH change. The result suggested that topiramate can rapidly induce an intracellular pH change in this pre-clinical tumor model. 

Zonisamide is a sulfonamide-type antiepileptic drug. It has been in use to treat epilepsy since at least 1990. In 2000, The Food and Drug Administration (FDA) approved it for use in the United States, suggesting that it be used as adjunctive or add-on therapy in the treatment of partial seizures in adults. In brain tumor patients, zonisamide and topiramate are among preferred drugs to prevent seizures. As far as we know, there are no major pre-clinical or clinical studies focusing on the effects of zonisamide in brain tumors. One experimental study has demonstrated that zonisamide has antiproliferative effects in C6 rat glioma cells [86]. 

Boyd and colleagues studied the use of SLC-0111, CA IX and CA XII inhibitor, in grade IV astrocytoma (glioblastoma) [77]. The current standard of care for glioblastoma is maximal surgical resection, followed by radiotherapy plus concomitant and maintenance TMZ. The authors used patient-derived glioblastoma multiforme xenograft cells and showed that SLC-0111, in combination but not without TMZ, decreased tumor growth in vitro. Furthermore, when these xenografts were treated with the combination of these two drugs, subcutaneous tumors significantly regressed compared to the drugs used alone. Experimental animals with orthotopic glioblastoma multiforme xenografts treated with the combination, survived longer. All mice in vehicle only and SLC-0111 groups died during the follow-up period of 130 days, whereas 2/5 mice in TMZ and 5/6 mice in combination groups survived until the end of the period. Therefore, the median survival was not calculated for the latter two groups. The authors stated that SCL-0111 should be studied in a clinical trial in combination with TMZ.

Wu and coworkers studied TMZ and molecular mediators for its clinical response in glioma patients with tumors that have a methylated O6-methylguanine DNA methyltransferase (MGMT) promoter [78]. They showed that tumors with high BCL-3, promoting a more malignant phenotype by inducing epithelial-to-mesenchymal transition, were associated with a poor response to TMZ. Furthermore, CA II was identified as a downstream factor that promoted chemoresistance. Finally, experiments in glioma xenograft mouse models (three different cell lines) showed that acetazolamide prolonged the survival of TMZ-treated mice. The obvious conclusion was that acetazolamide might be a potential drug as a chemosensitizer for treating TMZ-resistant gliomas. 

Xu and colleagues recently demonstrated the effect of another CA IX inhibitor, U-104, on glioma cells [87]. They showed that U-104 combined with TMZ efficiently induced cell death more than TMZ alone. Their findings introduced a SOX9/CA IX-mediated oncogenic pathway, the inhibition of which by a CA IX inhibitor can sensitize glioma cells to TMZ treatment. 

The role of CA XII in brain tumors is only a marginally explored research area. Salaroglio et al. showed that CA XII and P-glycoprotein coexpression is a new hallmark of chemoresistance in glioblastoma neurospheres, representing a previously unknown mechanism of TMZ resistance [88]. In another study, a psammaplin C derivative, a subnanomolar CA XII inhibitor, was found effective in-patient derived cell and xenograft models of glioblastoma [89]. The results showed that targeting P-glycoprotein/CA XII interaction may improve TMZ efficacy in vivo. The combination therapy with Psammaplin C derivative and TMZ was effective in restoring TMZ efficacy in vivo and increased overall survival of mice with TMZ-resistant tumors. 6-triazolyl-substituted sulfocoumarins have also been introduced as potential drugs to inhibit CA XII and affect P-glycoprotein activity [90]. One of these coumarin derivatives was able to sensitize multidrug-resistant non-small cell lung carcinoma cells to doxorubicin.

The finding that CA inhibitors have an effect in vivo when used in combination with TMZ could be partly explained by the evidence, that TMZ may also contribute to normalization of an acidic extracellular pH as shown in rat glioma models [91]. In other words, the combination of these drugs could effectively diminish glioma invasion and tumor growth by a dual mechanism i.e. by inhibiting extracellular acidification and also by disrupting physiological intracellular pH regulation. 

CA IX has been reported as a potential target for chimeric antigen receptor (CAR) T cell therapy against glioblastoma [92]. A 20% cure rate was found with anti-CA IX CAR T cells in a mouse U251 glioblastoma cell model.

### 4.2. Clinical Trials

CA inhibitors have been used for several decades for various clinical indications. There is already a vast clinical experience (especially from the use of pan-CA inhibitors like acetazolamide). In some cases, drug designers did not initially aim to develop CA inhibitors, but it was only later recognized that the novel compounds were indeed efficient CA inhibitors. This was the case e.g. with the anti-epileptic drug, topiramate [71,72,73,74,75,76,77,78].

Maschio and colleagues have reported a prospective observational study on 48 patients with brain metastases and epilepsy [93]. They tested efficacy, safety, and impact on life expectancy of levetiracetam, oxcarbazepine, and topiramate monotherapies in these patients. All these drugs showed favorable responses on epilepsy by reducing seizure frequency and producing only few side effects. Unfortunately, they failed to improve the survival of patients. 

There is currently an ongoing clinical trial (NCT03011671) for the use of a CA inhibitor in the therapy of malignant astrocytoma (MGMT methylated grade III-IV astrocytomas). The aim is to study the dose-limiting side effects of acetazolamide used in combination with TMZ. The study population includes patients who undergo treatment with standard adjuvant TMZ after completing treatment with TMZ and ionizing radiation. Unfortunately, there are no published results yet available. As a pan-CA inhibitor, acetazolamide can inhibit all human CAs, except the low activity CA III, at 2.5–250 nanomolar concentrations [3]. In fact, it is the most efficient inhibitor of CA VII which participates in neuronal signal transduction at least in hippocampal neurons [16]. Even though acetazolamide has been clinically used for decades and known as a relatively well-tolerated drug, it may also induce brain-related side-effects due to its unspecific mode of action. 

There is another highly interesting clinical trial (NCT02770378) underway to investigate the effect of a panel of nine repurposed drugs, including a CA inhibitor, celecoxib, in combination with TMZ for patients with recurrent glioblastoma. The previous published CUSP9* protocol involved the following drugs in addition to TMZ: aprepitant, artesunate, auranofin, captopril, celecoxib, disulfiram, itraconazole, ritonavir, and sertraline [94]. The clinical trial based on the present version 3 of the protocol, CUSP9v3, will be soon completed and reported. The newest version includes the same panel of drugs except artesunate that has been replaced with minocycline. Notably, one of the drugs is celecoxib which is a well-known CA inhibitor [95]. In addition to brain tumors, there are ongoing clinical trials on the combination of acetazolamide, radiotherapy, chemotherapy with platinum and etoposide in localized small cell lung cancer (https://clinicaltrials.gov/ct2/show/NCT03467360), and on combination of SLC-0111 and gemcitabine in metastatic pancreatic ductal cancer (https://clinicaltrials.gov/ct2/show/NCT03450018).

## 5. Conclusions

Brain tumors cause significant morbidity and mortality, and thus novel treatment modalities are needed. Overexpression of CAs has been shown in several brain tumors, and importantly CA II, CA IX, and CA XII can be used to determine patient prognosis in certain tumor categories. There are already several in vitro and in vivo studies describing the potential benefit of CA inhibitors in treatment of cancer, and few of those show promising results also in preclinical trials of brain tumors, especially in combination with other cytotoxic drugs. Nevertheless, we are still lacking large clinical trials and data about their potential value in the treatment of brain cancer.

## Figures and Tables

**Figure 1 cancers-12-01723-f001:**
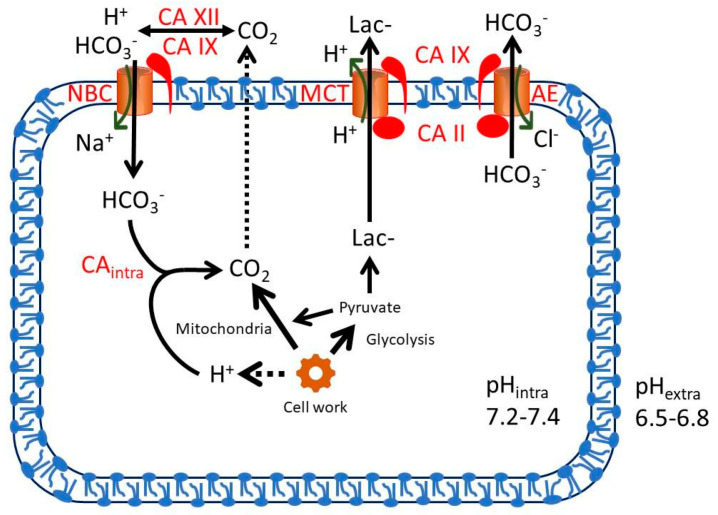
Schematic representation of tumor pH regulation focusing on Carbonic anhydrases (CAs). Regulation is based on the interplay between CAs and transporter proteins. Both glycolysis and mitochondrial respiration produce metabolic acids. Anaerobic glycolysis yields lactate and protons that are excreted from the cell by monocarboxylate transporters (MCTs). Aerobic respiration produces CO_2_, which is hydrated by the CA-catalyzed reaction. CO_2_ can exit the cell by passive diffusion across the plasma membrane. Efficient pH regulation requires several transporters and enzymes exporting protons from the cell or transporting HCO_3_^−^ (including Na+/HCO3 cotransporters (NBCs) and Cl−/HCO3 exchanger (AEs)) and some of them, such as Na^+^/H^+^ exchanger 1 and vacuolar H^+^-ATPase, are not shown in this figure. Elimination of CO_2_ is supported by extracellular CA IX and CA XII and intracellular CA II (CAintra). Transport activities are facilitated by metabolon systems involving both CAs and transport proteins. The figure has been modified from [10].

**Figure 2 cancers-12-01723-f002:**
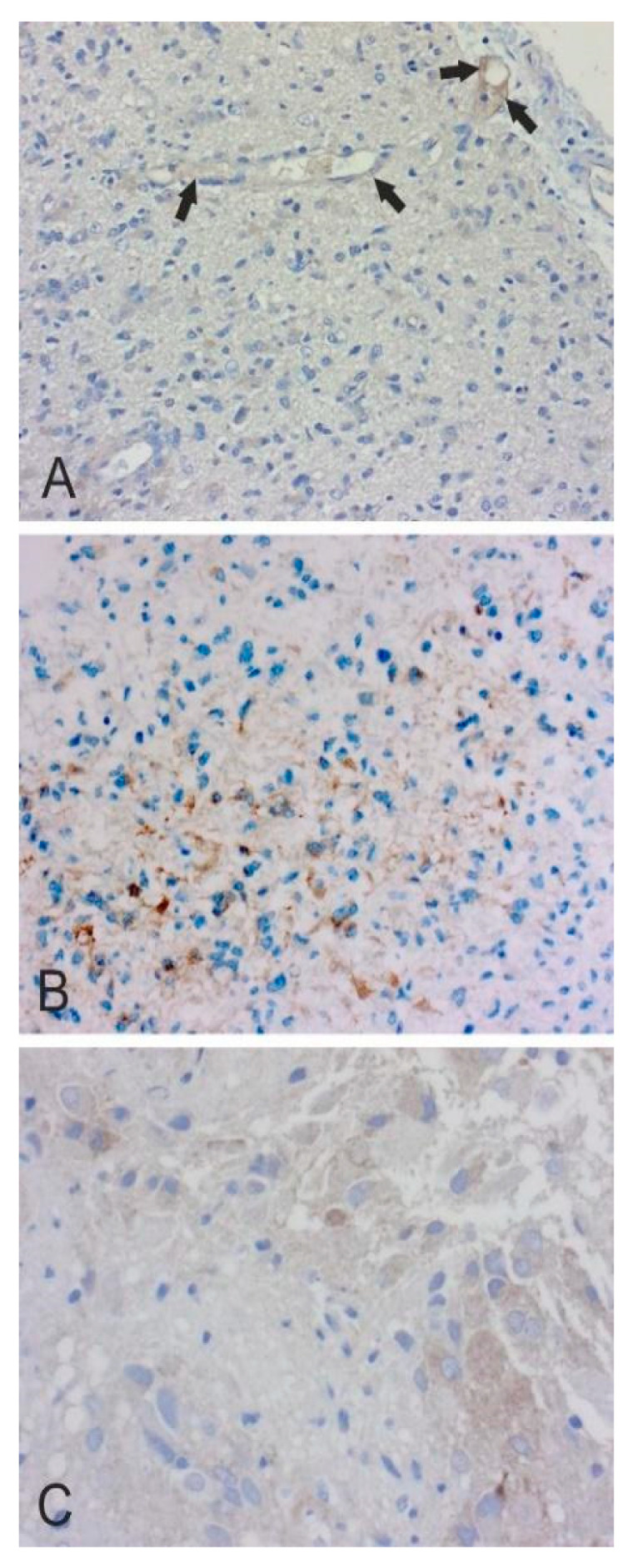
(**A**). Weak endothelial CA II staining (arrows) in small capillaries of grade III anaplastic astrocytoma (×200). (**B**). Faint cytoplasmic CA IX staining in grade II diffuse astrocytoma (×200). (**C**). Faint CA XII staining in grade III anaplastic astrocytoma (×400).

**Figure 3 cancers-12-01723-f003:**
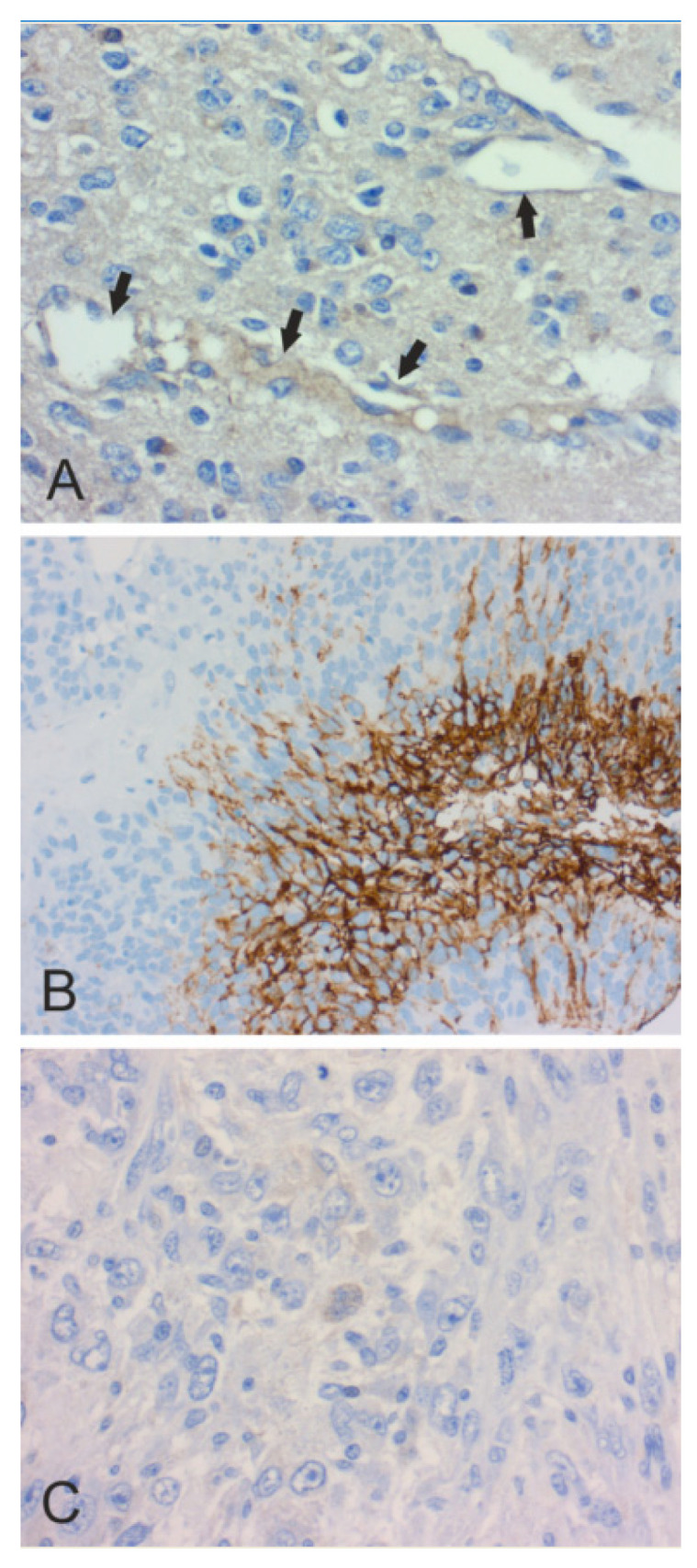
Immunohistochemical staining of CA in grade IV glioblastoma. (**A**). Moderate endothelial CA II staining (arrows) (×400). Tumor cells are very weakly CA II-positive. (**B**). Strong perinecrotic CA IX staining (×200). (**C**). Faintly CA XII-positive multinucleated tumor cell in the middle (×400).

**Figure 4 cancers-12-01723-f004:**
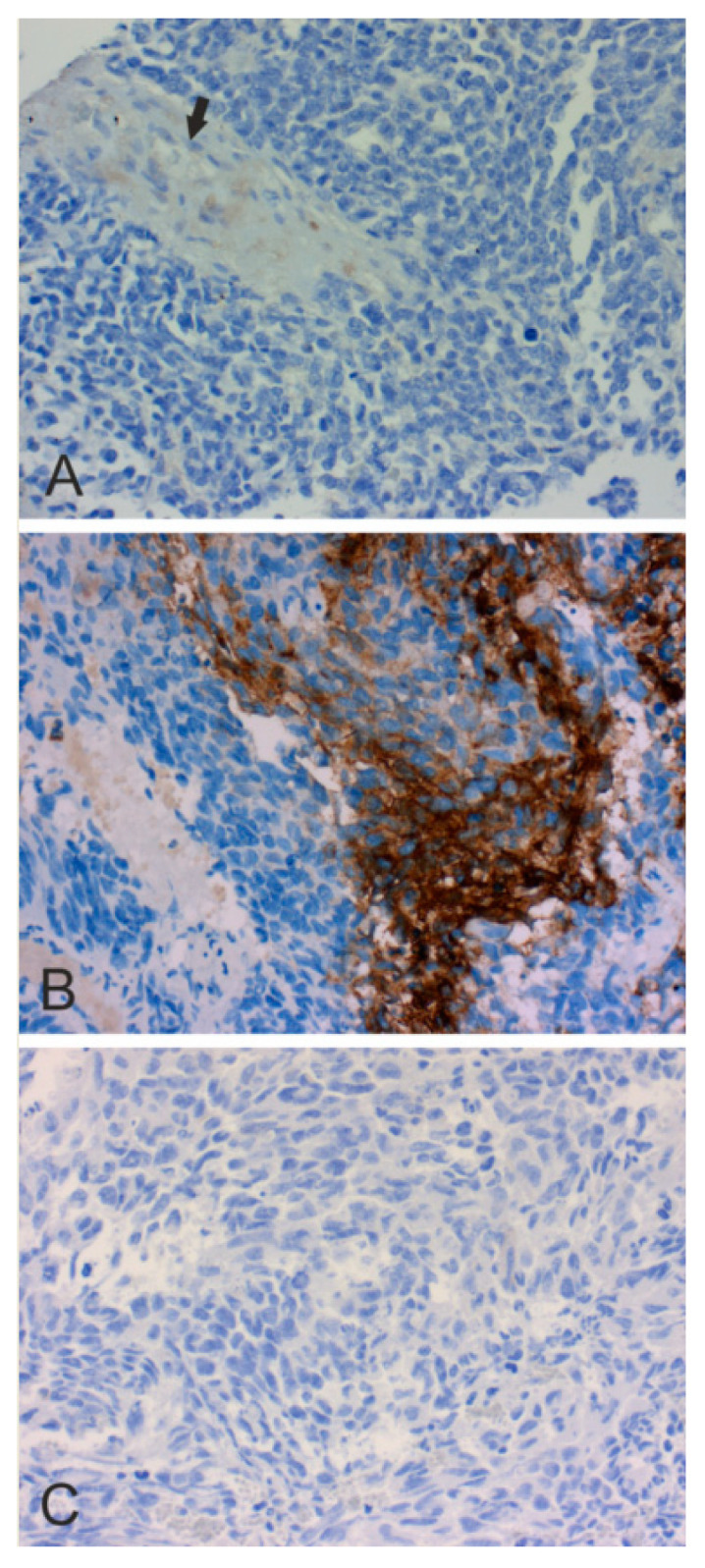
CAs II, IX and XII in medulloblastoma. (**A**). Faint immunohistochemical CA II staining in a thickened vessel wall (arrow) (×200). (**B**). Strong CA IX staining (×400). (**C**). Negative CA XII staining (×200).

**Figure 5 cancers-12-01723-f005:**
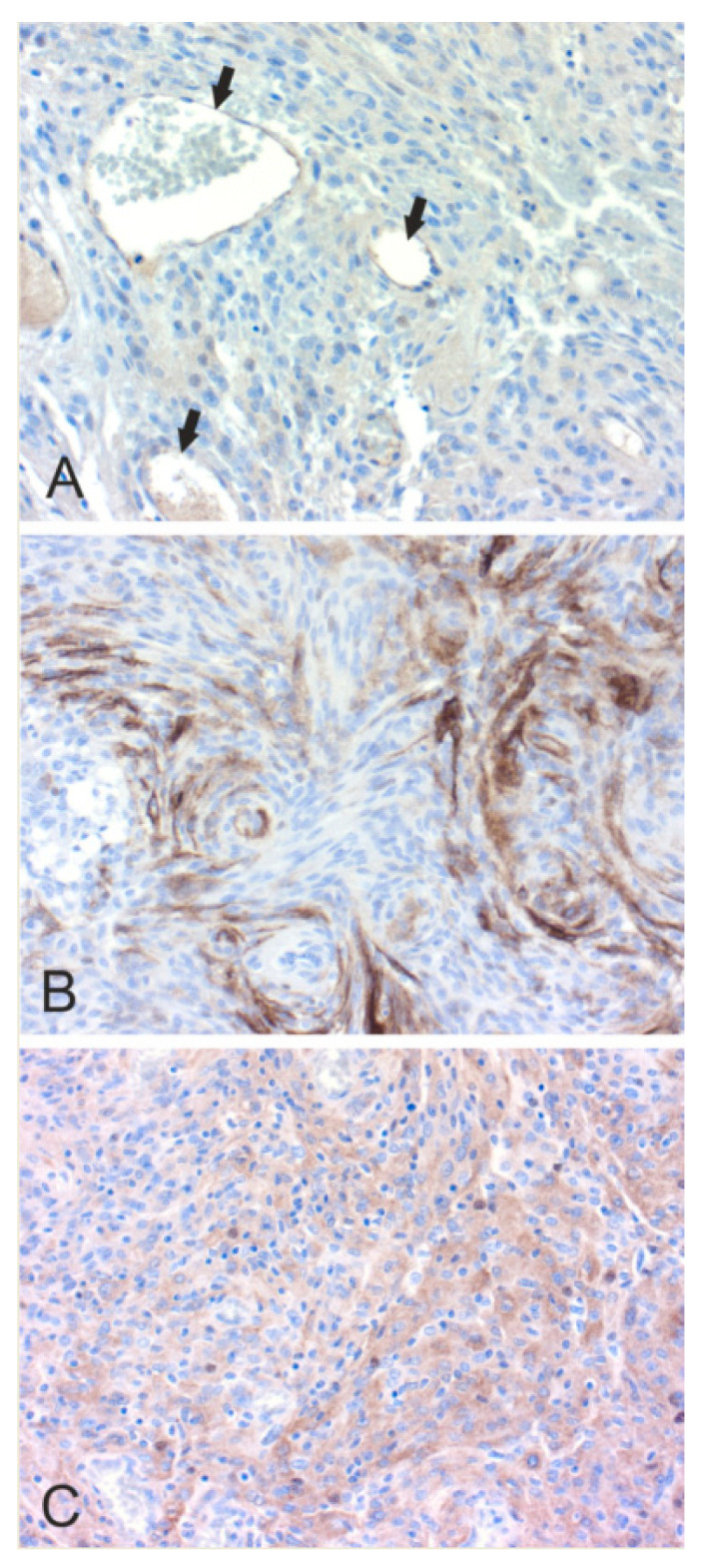
Immunohistochemical staining of CA II, CA IX and CA XII in meningiomas. (**A**). Faint endothelial CA II staining (×200). (**B**). Strong cytoplasmic staining for CA IX (×200). (**C**). Faint to moderate staining for CA XII (×200).

**Figure 6 cancers-12-01723-f006:**
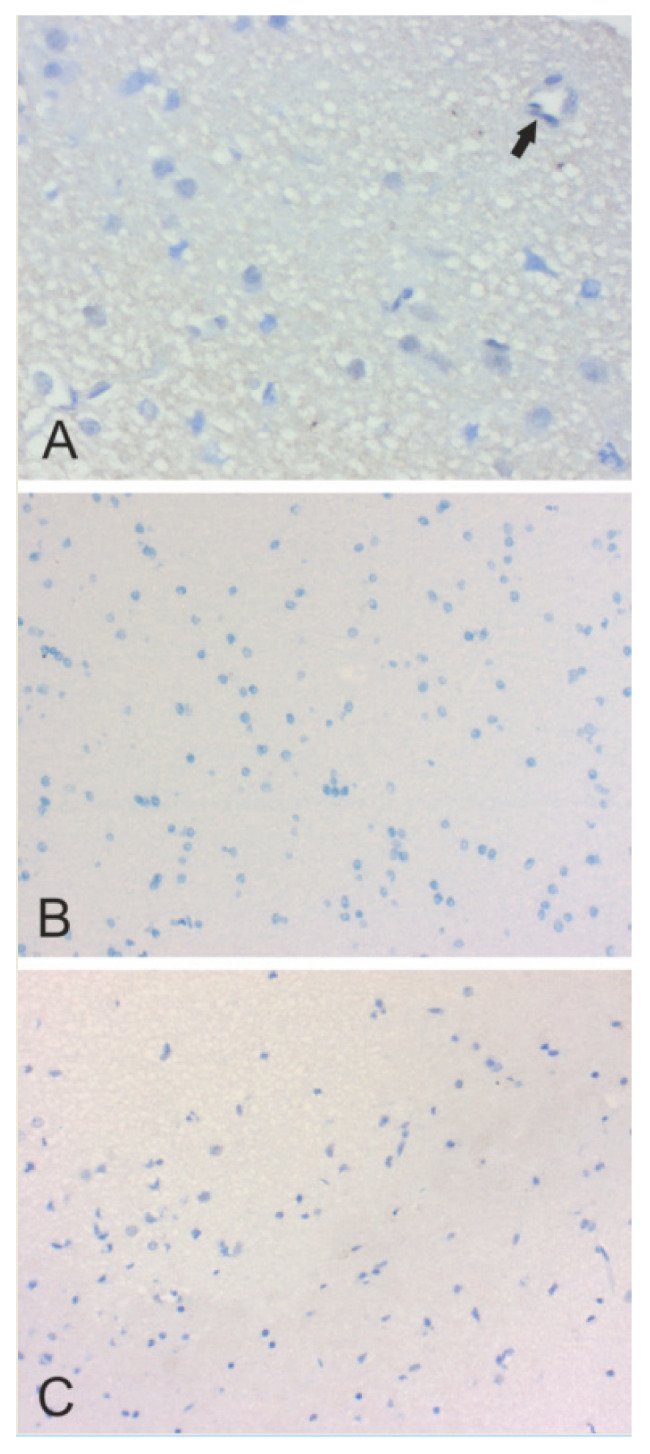
Immunohistochemical staining of CA II (×400) (**A**), CA IX (×200) (**B**) and CA XII (×200) (**C**) in the normal human brain. The immunostained CA IX and CA XII remained negative in these sections. CA II shows weak punctate signal, probably indicating a myelin-associated reaction. A negative capillary is marked with an arrow.

**Table 1 cancers-12-01723-t001:** Expression of CAs in different cell types of the CNS.

Cell Type	CA Isozyme
Astrocytes	II, V [15,27,28,32,33,34,35,36]
Choroid plexus	II, III, XII, XIV [21,37,38,39,40,41]
Endothelial cells	IV [42]
Microglial cells	III [31]
Myelin sheath	II [21,23]
Neurons	II, V, VII, XIV [16,32,37,39]
Oligodendrocytes	II, XIII [21,23,24,25,26,37,43]

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
