# Peer review of "The Expression of Carbonic Anhydrases II, IX and XII in Brain Tumors"

_cancers, 2020, doi:10.3390/cancers12071723_

Round 1
Reviewer 1 Report
It is a well written review regarding carbonic anydrases (CA) use in
predicting prognosis of brain tumor patients. Already completed and ongoing studies focusing on CA inhibition as a potential anti-cancer strategy are also included in this manuscript.
You could add more details on preclinical evidence for a treatment effect of acetazolamide and other antitumor drugs combinations. Which agents seem as most promising to add in a combination treatment protocol with CA and why? Which is the pathophysiologic mechanism?
Author Response
Reviewer 1
It is a well written review regarding carbonic anhydrases (CA) use in predicting prognosis of brain tumor patients. Already completed and ongoing studies focusing on CA inhibition as a potential anti-cancer strategy are also included in this manuscript.
You could add more details on preclinical evidence for a treatment effect of acetazolamide and other antitumor drugs combinations. Which agents seem as most promising to add in a combination treatment protocol with CA and why? Which is the pathophysiologic mechanism?
Our comment: Based on the reviewer´s comment we have added several examples to sections “pre-clinical studies” and “clinical trials” where the classical drug, acetazolamide, anticonvulsants, topiramate and zonisamide, and a CA IX inhibitor, SLC-0111, have been tested and described. (lines 289-333, 341-344, 354-376, and 406-413)
Reviewer 2 Report
This is a well written informative review manuscript on CAs in brain tumors. CAs and in particular CAs II, IX and XII are relevant for these tumors as is also shown by the preclinical studies and clinical trial of CA inhibitors in combination with TMZ chemotherapy. Therefore, I consider this manuscript as a potential important contribution to our understanding of brain tumors.
I have a number of suggestions to revise the manuscript.
- Abstract. Well written. But it should become clear here what type of brain tumors. Secondary brain tumors (metastases) are only mentioned very superficially in lines 108-109.
- Introduction. Excellent description of CAs
- In section 2.2, it is not clear when animal models are discussed and when human brain tumor. Moreover, it is not made clear whether the role of CAs in mice and rats is the same as in patients. What I worked out myself is that lines 56-70 including Table 1 is about rats and mice and then from line 72 onwards is about human. This should be more clear. Moreover, I miss a (semi-)quantitative analysis in Table 1. It is stated that the choroid plexus is one of the few sites in CNS where CAs are expressed which is now contradictory with Table 1.
- A table with expression patterns in human brain and in the various types of brain cancer would be very helpful.
- Fig 1 and 2. Controls are not shown or mentioned. I would like to know whether the staining is specific. What types of cells are strongly positive in Fig 1B and which cell types are negative? In Fig 2A blood vessels are positive. This looks to me that not only ECs are positive but also cells or ECM adjacent to ECs. Furthermore, both in Fig 1A and B I see a lot of nuclear staining rather than cytoplasmic staining. This surprises me (is this non-specific staining?). Explanation?
- What surprises me as well is that both CA inhibitors show effect when they are adjuvant to TMZ treatment. This may well be an important issue to improve our understanding of the role of CAs in brain tumors. I would appreciate a discussion on this phenomenon.
I hope that my comments will be useful to further improve the quality of the manuscript.
Author Response
Reviewer 2
This is a well written informative review manuscript on CAs in brain tumors. CAs and in particular CAs II, IX and XII are relevant for these tumors as is also shown by the preclinical studies and clinical trial of CA inhibitors in combination with TMZ chemotherapy. Therefore, I consider this manuscript as a potential important contribution to our understanding of brain tumors.
I have a number of suggestions to revise the manuscript.
- Abstract. Well written. But it should become clear here what type of brain tumors. Secondary brain tumors (metastases) are only mentioned very superficially.
Our comment 1: We have modified the last part of the abstract as follows: “We review here their role in tumorigenesis and potential value in predicting prognosis of brain tumors, including astrocytomas, oligodendrogliomas, ependymomas, medulloblastomas, meningiomas, and craniopharyngiomas. We also introduce both already completed and ongoing studies focusing on CA inhibition as a potential anti-cancer strategy.” (lines 23-26)
Our comment 2: We agree that metastases were mentioned superficially in the previous version. In fact, there are not many reports available to date. We have added a new paragraph describing the data published on the expression of CA IX in brain metastases. (lines 272-284)
- Introduction. Excellent description of CAs
Our comment: We thank the reviewer for this positive comment.
- In section 2.2, it is not clear when animal models are discussed and when human brain tumor. Moreover, it is not made clear whether the role of CAs in mice and rats is the same as in patients. What I worked out myself is that lines 56-70 including Table 1 is about rats and mice and then from line 72 onwards is about human. This should be more clear. Moreover, I miss a (semi-)quantitative analysis in Table 1. It is stated that the choroid plexus is one of the few sites in CNS where CAs are expressed which is now contradictory with Table 1.
Our comment: We thank the reviewer for pointing out this issue. We have clarified the text by adding new titles:
Added a title: 2.2.1 CA expression based on animal models (line 71)
Added a title: 2.2.2 CA expression in human brain tissue (line 90)
We also deleted the word “few” and the new sentence reads: “The choroid plexus is one of the sites in CNS where CAs are expressed” (line 72)
- A table with expression patterns in human brain and in the various types of brain cancer would be very helpful.
Our comment: We show Table 1 with information about CAs in normal brain cells. When building this Table we realized that it is rather complicated, because immunostainings have been performed with different antibodies and staining methods during several decades. This same phenomenon would make it very difficult to design a new Table with reliable data on various tumors – especially as the classification criteria of brain tumors have been changed during the last years. For the future we would prefer RNASeq-type analyses of large datasets with up-to-date tumor classification. This kind of analysis would make an interesting original research paper.
- Fig 1 and 2. Controls are not shown or mentioned. I would like to know whether the staining is specific. What types of cells are strongly positive in Fig 1B and which cell types are negative? In Fig 2A blood vessels are positive. This looks to me that not only ECs are positive but also cells or ECM adjacent to ECs. Furthermore, both in Fig 1A and B I see a lot of nuclear staining rather than cytoplasmic staining. This surprises me (is this non-specific staining?). Explanation?
Our comment: We thank the reviewer for this valuable comment. According to both this and editor´s comment we have deleted previous Figure 1 and 2 and prepared new immunohistochemical staining images for CA II, CA IX and CA XII in the normal brain and astrocytoma, glioblastoma, meningioma and medulloblastoma specimens (Figures 2-6). Description of the figures has been added (lines 128-132) as well as figure legends (lines 762-780)
- What surprises me as well is that both CA inhibitors show effect when they are adjuvant to TMZ treatment. This may well be an important issue to improve our understanding of the role of CAs in brain tumors. I would appreciate a discussion on this phenomenon.
Our comment: We have added the following sentences: “The finding that CA inhibitors have an effect in vivo when used in combination with TMZ could be partly explained by the evidence, that TMZ may also contribute to normalization of an acidic extracellular pH as shown in rat glioma models (90). In other words, the combination of these drugs could effectively diminish glioma invasion and tumor growth by a dual mechanism i.e. by inhibiting extracellular acidification and also by disrupting physiological intracellular pH regulation.” (lines 369-373)
Reviewer 3 Report
In general, this ms is well written in modern scientific/medical English usage. Exceptions noted below. This is a didactic piece, lightly reviewing selected aspects of CAII and IX in gliomas. In a work like that a Table with definition, histological description, and cell origin, listing the various gliomas will be required. A rewrite using modern, glioma WHO terminology uniformly throughout the ms. is required.
Line 30. It might be better to add “one of the elements that regulate…” or “CAs participate in regulation of…” since there are several (many ?) physiological paths by which intra- and extra- cellular pH is regulated.
There are 2 commonly clinically used CA inhibitors the authors should discuss - topiramate and zonisamide. A third commonly used CA II and CA IX is currently in a clinical trial in treatment of human glioblastoma, NCT 02770378 must be discussed, particularly as this clinical study has reported preliminary positive results.
Not quoting the paper of Albatany et al is intolerable in a paper on CA in glioblastomas, and not quoting Amiri et al indicates either dishonesty or sloppiness.
Line 50, do the authors intend to say “...extracellular fluid…” or what fluid ?
Line 56 is wrong and must be corrected. Glia and neurons and elements of brain vasculature all express various CA isoforms as the authors partially say later in the ms. and their Table 1 [which was a helpful addition to the text].
Figure 1 is superb ! However the authors’ use of figures is disorganized and must be corrected. In a didactic piece as this ms. figures and their clarity are particularly important. Keeping the micrographs large as they now are is crucially important and eminently useful. Fig 2 is a bit odd and less useful. Showing 2 different stains on two different types of tumor doesn’t allow readers to compare tumors. May I suggest reorganizing the micrographs, showing CAII and CAIX in fig.1 glioblastoma, then another, fig. 2, CAII and CAIX in grade III glioma, then fig. 3, CAII and CAIX in medulloblastoma, then fig. 4 CAII and CAIX in meningioma ?
The authors shift back and forth between using the word “astrocytoma” and “glioma”. Better would be sticking to the current standard of using WHO terminology, glioma, then specifying grade [I, II, III, or IV]. Or simply “glioma” if NOS. The authors’ confused use of these terms
leads to several incorrect and self-contradictory statements. The terminology used in this ms. must be corrected and made uniform. Also when mentioning glioblastoma the authors must either put a comma between “grade IV glioma” and glioblastoma or preferably say simply glioblastoma. All glioblastomas are grade IV and all grade IV gliomas are glioblastomas.
Lines 162-164, is incorrect. Age was also a prognostic factor in both.
Lines 170-174 need revision. “Interestingly, the study population consisted of recurrent malignant glioma patients with 171 measurable recurrent or residual primary disease on contrast-enhanced MRI or CT scan, and already given chemotherapy and possible radiotherapy, resembling tumors with possible clonal selection by previous cancer treatments.” if the underlined phrase makes any sense to others it might be my misunderstanding only. I don’t understand what the authors wish to say there.
Lines 154 through 185, when enumerating as the authors do, it is easier for readers to follow if “First…”, and “Second…” etc are each given their own paragraph.
Obviously lines 174-175 need fixing. “Third, Yoo and co-workers investigated 78 WHO grade II, III, and IV astrocytic gliomas by immunohistochemistry, and found that CA IX expression was associated with higher-grade histology and worse survival in the whole patient cohort and WHO grade II tumors. “ The authors might know what they want to say but others won’t.
If the authors quote Flynn et al and Proecholdt et al the must then discuss these discrepant results. If the authors cannot find meaningful elements to account for the different, discordant findings they must seek out glioblastoma professionals who can explain the origin of discrepancies to them.
The authors cannot say “As a conclusion from the articles presented above, CA IX expression is a common phenomenon in 182 malignant astrocytic gliomas, and it can be used as a prognosticator in grade II-IV diffusely 183 infiltrating astrocytomas, and separately in glioblastomas. “ implying a blanket disregard for Flynn et al. They may argue why results of Flynn et al are inaccurate but they must present that argument to us.
In general presenting CAII and IX as prognostic indices is fairly trivial, and as in above contested data anyway. A prognosis matters little, dying in 18 months or 20 months as far as treatment, or a patient’s planning. Look at the s.d. Behind those figures. CA expression as a treatment target matters tremendously. I recommend the authors rewrite this paper, eliminating or at least de-emphasizing prognostic value of CAII-IX and increase their focus on treatment implications. We have many marketed CAII-IX-XII inhibitors that all clinicians are familiar with.
Acetazolamide, celecoxib, topiramate and zonisamide are 4 of these. If the authors insisted on retaining their focus on prognosis implications of CA IHC findings, I personally would reject the paper.
Line 261, it is customary to list the NCT number of clinical trials.
I don’t understand why the authors have not discussed the preclinical & clinical work on using already-marketed CAII-IX inhibitors in gliomas, references 1 through 8 below. As mentioned NCT 02770378 has already reported preliminary positive findings in recurrent glioblastoma. These findings [1-8 below] must be part of any discussion of CA function in gliomas.
In a didactic piece like this a figure [a schematic or “cartoon”] showing how CA regulated extra- and intra- cellular pH must be given.
The authors discuss their own work [Neuro Oncol. 2008 Apr; 10(2): 131–138.] but I didnt understand that work then, nor do I on re-reading now. Fig. 3 in that 2008 work seems to show CA IHC staining having marginal but some influence on OS but what exactly was the glioma studied ? What are diffusely infiltrating astrocytomas ? Did they or did they not include grade III and grade IV tumors ? Or...?
As mentioned above the authors’ discussion of CAII/CAIX inhibitors [starting line 228] in past work is grossly and intolerable inadequate.
________________________________________
Albatany M, Ostapchenko VG, Meakin S, Bartha R. Brain tumor acidification using drugs simultaneously targeting multiple pH regulatory mechanisms. J Neurooncol. 2019 Sep;144(3):453-462. doi: 10.1007/s11060-019-03251-7.
Amiri A, Le PU, Moquin A, Machkalyan G, Petrecca K, Gillard JW, Yoganathan N, Maysinger D. Inhibition of carbonic anhydrase IX in glioblastoma multiforme. Eur J Pharm Biopharm. 2016 Dec;109:81-92. doi: 10.1016/j.ejpb.2016.09.018.
Rubin RC, Henderson ES, Ommaya AK, Walker MD, Rall DP. The production of cerebrospinal fluid in man and its modification by acetazolamide. J Neurosurg. 1966 Oct;25(4):430-6. doi: 10.3171/jns.1966.25.4.0430.
- Cui J, Zhang Q, Song Q, Wang H, Dmitriev P, Sun MY, Cao X, Wang Y, Guo L, Indig IH, Rosenblum JS, Ji C, Cao D, Yang K, Gilbert MR, Yao Y, Zhuang Z.Targeting hypoxia downstream signaling protein, CAIX, for CAR T-cell therapy against glioblastoma. Neuro Oncol. 2019 Nov 4;21(11):1436-1446. doi:10.1093/neuonc/noz117.
- Podolski-Renić A, Dinić J, Stanković T, Jovanović M, Ramović A, Pustenko A, Žalubovskis R, Pešić M. Sulfocoumarins, specific carbonic anhydrase IX and XII inhibitors, interact with cancer multidrug resistant phenotype through pH regulation and reverse P-glycoprotein mediated resistance. Eur J Pharm Sci. 2019Oct 1;138:105012. doi: 10.1016/j.ejps.2019.105012.
- Hannen R, Selmansberger M, Hauswald M, Pagenstecher A, Nist A, Stiewe T, Acker T, Carl B, Nimsky C, Bartsch JW. Comparative Transcriptomic Analysis of Temozolomide Resistant Primary GBM Stem-Like Cells and Recurrent GBM Identifies Up-Regulation of the Carbonic Anhydrase <i>CA2</i> Gene as Resistance Factor. Cancers (Basel). 2019 Jun 30;11(7):921. doi: 10.3390/cancers11070921.
- Mujumdar P, Kopecka J, Bua S, Supuran CT, Riganti C, Poulsen SA. Carbonic Anhydrase XII Inhibitors Overcome Temozolomide Resistance in Glioblastoma. J Med Chem. 2019 Apr 25;62(8):4174-4192. doi: 10.1021/acs.jmedchem.9b00282.
- Salaroglio IC, Mujumdar P, Annovazzi L, Kopecka J, Mellai M, Schiffer D, Poulsen SA, Riganti C. Carbonic Anhydrase XII Inhibitors OvercomeP-Glycoprotein-Mediated Resistance to Temozolomide in Glioblastoma. Mol Cancer Ther. 2018 Dec;17(12):2598-2609. doi: 10.1158/1535-7163.MCT-18-0533.
- Cetin B, Gonul II, Gumusay O, Bilgetekin I, Algin E, Ozet A, Uner A. Carbonic anhydrase IX is a prognostic biomarker in glioblastoma multiforme. Neuropathology. 2018 Oct;38(5):457-462. doi: 10.1111/neup.12485.
- Xu X, Wang Z, Liu N, Cheng Y, Jin W, Zhang P, Wang X, Yang H, Liu H, Zhang Y, Tu Y. Association between SOX9 and CA9 in glioma, and its effects on chemosensitivity to TMZ. Int J Oncol. 2018 Jul;53(1):189-202. doi:10.3892/ijo.2018.4382.
- Kast RE, Karpel-Massler G, Halatsch ME. CUSP9* treatment protocol for recurrent glioblastoma: aprepitant, artesunate, auranofin, captopril, celecoxib, disulfiram, itraconazole, ritonavir, sertraline augmenting continuous low dose temozolomide. Oncotarget. 2014 Sep 30;5(18):8052-82. doi:10.18632/oncotarget.2408.
Author Response
Reviewer 3
In general, this ms is well written in modern scientific/medical English usage. Exceptions noted below. This is a didactic piece, lightly reviewing selected aspects of CAII and IX in gliomas. In a work like that a Table with definition, histological description, and cell origin, listing the various gliomas will be required. A rewrite using modern, glioma WHO terminology uniformly throughout the ms. is required.
Line 30. It might be better to add “one of the elements that regulate…” or “CAs participate in regulation of…” since there are several (many ?) physiological paths by which intra- and extra- cellular pH is regulated.
Our comment 1: We show Table 1 with information from normal brain cells. When building this Table we realized that it is rather complicated, because immunostainings have been performed with different antibodies and staining methods during several decades. This same phenomenon makes it difficult to design a new Table with reliable data on various tumors – especially as the classification criteria of brain tumors have been changed. For the future we would prefer RNASeq-type analyses of large datasets with up-to-date tumor classification. This kind of analysis would make an interesting original research paper.
Our comment 2: The manuscript now follows terminology guidelines of “WHO 2016 Classification of tumors of the central nervous system” wherever cited to our own studies. We think that it is fair to use original terminology when citing previous articles published by others. Changing all tumor types to WHO 2016 based definitions is not feasible, because it is not possible to gather all needed molecular information from previous studies, especially from other research groups. E.g. IDH mutation status was reported to be an important feature in 2008 (Parsons et al Science 2008 Sep 26;321(5897):1807-12).
Our comment 3: We have corrected the sentence pointed out by the reviewer. “CAs participate in various physiological roles, such as regulation of intra-and extracellular pH and ion transport, as well as water and electrolyte balance.” (lines 37-39)
There are 2 commonly clinically used CA inhibitors the authors should discuss - topiramate and zonisamide. A third commonly used CA II and CA IX is currently in a clinical trial in treatment of human glioblastoma, NCT 02770378 must be discussed, particularly as this clinical study has reported preliminary positive results.
Our comment: We have now included discussion about topiramate and zonisamide (lines 289-290, 314-333) and also discussed the clinical trial pointed out by the reviewer (lines 406-413).
Not quoting the paper of Albatany et al is intolerable in a paper on CA in glioblastomas, and not quoting Amiri et al indicates either dishonesty or sloppiness.
Our comment: Both Albatany et al. and Amiri et al. papers are now cited (lines 304-309 and 298-303)
Line 50, do the authors intend to say “...extracellular fluid…” or what fluid ?
Our comment: extracellular fluid (line 64)
Line 56 is wrong and must be corrected. Glia and neurons and elements of brain vasculature all express various CA isoforms as the authors partially say later in the ms. and their Table 1 [which was a helpful addition to the text].
Our comment: We deleted the word “few”, which might have been misleading. The corrected sentence reads: “The choroid plexus is one of the sites in CNS where CAs are expressed.” (line 72)
Figure 1 is superb! However the authors’ use of figures is disorganized and must be corrected. In a didactic piece as this ms. figures and their clarity are particularly important. Keeping the micrographs large as they now are is crucially important and eminently useful. Fig 2 is a bit odd and less useful. Showing 2 different stains on two different types of tumor doesn’t allow readers to compare tumors.
May I suggest reorganizing the micrographs, showing CAII and CAIX in fig.1 glioblastoma, then another, fig. 2, CAII and CAIX in grade III glioma, then fig. 3, CAII and CAIX in medulloblastoma, then fig. 4 CAII and CAIX in meningioma?
Comment: Immunohistochemical figures have been replaced. According to the editor´s and reviewers´suggestions we have deleted previous Figures 1 and 2 and prepared new immunohistochemical staining images for CA II, CA IX and CA XII in the normal brain and astrocytoma, glioblastoma, meningioma and medulloblastoma (Figures 2-6). Description of the figures has been added (lines 128-132) as well as figure legends (lines 762-780)
The authors shift back and forth between using the word “astrocytoma” and “glioma”. Better would be sticking to the current standard of using WHO terminology, glioma, then specifying grade [I, II, III, or IV]. Or simply “glioma” if NOS. The authors’ confused use of these terms leads to several incorrect and self-contradictory statements. The terminology used in this ms. must be corrected and made uniform. Also when mentioning glioblastoma the authors must either put a comma between “grade IV glioma” and glioblastoma or preferably say simply glioblastoma. All glioblastomas are grade IV and all grade IV gliomas are glioblastomas.
Our comment: The manuscript now follows terminology guidelines of “WHO 2016 Classification of tumors of the central nervous system” wherever cited to our own studies. We think that it is fair to use original terminology when citing previous articles published by others. Changing all tumor types to WHO 2016 based definitions is not feasible, since it is not possible to gather all needed molecular information from previous studies, especially from other research groups. E.g. IDH mutation status was reported to be an important feature in 2008 (Parsons et al Science 2008 Sep 26;321(5897):1807-12).
Lines 162-164, is incorrect. Age was also a prognostic factor in both.
Our comment: The cited paper states: “Multivariate analysis in the entire cohort selected the age of the patient and the interaction model of CAIX with grade, HIF-1a, VEGF, and microvascular parameters losing significance in this regard. When multivariate analysis was restricted to grades II/III, CAIX levels emerged as the only factor influencing postoperative survival.”
We slightly modified the first sentence (lines 179-180)
Lines 170-174 need revision. “Interestingly, the study population consisted of recurrent malignant glioma patients with 171 measurable recurrent or residual primary disease on contrast-enhanced MRI or CT scan, and already given chemotherapy and possible radiotherapy, resembling tumors with possible clonal selection by previous cancer treatments.” if the underlined phrase makes any sense to others it might be my misunderstanding only. I don’t understand what the authors wish to say there.
Our comment: We modified the sentence as follows: “The study population consisted of recurrent malignant glioma patients with measurable recurrent or residual primary disease, already treated with chemotherapy with or without radiotherapy. Thus, the patients had tumors affected by cancer treatment and these recurrent tumors could be considered a model to study clonal selection. Again, positive CA IX immunostaining was associated with poor survival outcome in both univariate and multivariate analyses.” (lines 187-193)
Lines 154 through 185, when enumerating as the authors do, it is easier for readers to follow if “First…”, and “Second…” etc are each given their own paragraph.
Our comment: Each statement are now placed in independent paragraphs as suggested. (lines: 177-220)
Obviously lines 174-175 need fixing. “Third, Yoo and co-workers investigated 78 WHO grade II, III, and IV astrocytic gliomas by immunohistochemistry, and found that CA IX expression was associated with higher-grade histology and worse survival in the whole patient cohort and WHO grade II tumors. “ The authors might know what they want to say but others won’t.
Our comment: We modified the sentence as follows: “Third, Yoo and co-workers investigated 78 WHO grade II, III, and IV astrocytic gliomas by immunohistochemistry, and found that CA IX expression was associated with increasing WHO grade and poor survival in grade II-IV tumors, as well as with poor survival in WHO grade II tumors when these tumors were evaluated separately (53).” (Lines 194-198)
If the authors quote Flynn et al and Proecholdt et al the must then discuss these discrepant results. If the authors cannot find meaningful elements to account for the different, discordant findings they must seek out glioblastoma professionals who can explain the origin of discrepancies to them. The authors cannot say “As a conclusion from the articles presented above, CA IX expression is a common phenomenon in 182 malignant astrocytic gliomas, and it can be used as a prognosticator in grade II-IV diffusely 183 infiltrating astrocytomas, and separately in glioblastomas.“ implying a blanket disregard for Flynn et al. They may argue why results of Flynn et al are inaccurate but they must present that argument to us.
Our comment: We sincerely think that there is no actual discrepancy. There are already several published papers listed in our manuscript that have proposed CA IX as a useful marker to predict prognosis in malignant astrocytic tumors. The only article, which did not find a significant p-value, even though they described a similar trend in Kaplan-Meier curves, was the article by Flynn et al. They used a different primary antibody and scoring system which may have affected the results.
We modified the sentences and added the following information: “Fourth, Flynn and colleagues evaluated hypoxia-regulated proteins, including CA IX, and survival with adult glioblastomas, but did not find any statistically significant association (54). Notably, they used a commercial antibody against CA IX instead of the well characterized M75 antibody which has been used by numerous investigators and shown as highly specific against CA IX (50,51). In their analysis, the immunohistochemical score for CA IX remained at relatively low level (a little over 2 on scale 0-4). This could be due to low reactivity of the antibody or specific scoring system (counting only the percentage of positive cells without taking into account the staining intensity). Even then the higher CA IX signal showed a slight, although non-significant, correlation with poorer survival.” (lines 199-206)
In general presenting CAII and IX as prognostic indices is fairly trivial, and as in above contested data anyway. A prognosis matters little, dying in 18 months or 20 months as far as treatment, or a patient’s planning. Look at the s.d. Behind those figures. CA expression as a treatment target matters tremendously. I recommend the authors rewrite this paper, eliminating or at least de-emphasizing prognostic value of CAII-IX and increase their focus on treatment implications. We have many marketed CAII-IX-XII inhibitors that all clinicians are familiar with.acetazolamide, celecoxib, topiramate and zonisamide are 4 of these. If the authors insisted on retaining their focus on prognosis implications of CA IHC findings, I personally would reject the paper.
Our comment 1: We agree that the potential use of CAs as target molecules for future therapeutic applications is very important. The authors of this manuscript have been involved in neuropathological diagnostics, neurosurgical treatment, and scientific research of brain tumors, and are indeed, anxiously waiting for the CA inhibitor trials to be completed. We have edited the manuscript and added more text about CA inhibitors and also in vitro data. (lines 289-333, 341-344, 354-376, 382-394, 400-413)
Our comment 2: Reviewer 3 states that “A prognosis matters little, dying in 18 months or 20 months as far as treatment, or a patient's planning.” We do not fully agree with this statement. There might be misunderstanding, since Proecholdt et al. showed that median survival time differed 19 months (GBM, high expression 15 months vs low expression 34 months). Sathornsumetee et al (2008) show that there is a nine months overall survival difference between the CA IX-positive and -negative GBMs. Furthermore, we added a new reference (Cetin et al. 2018), and according to their results from GBM patients, the patients with CA overexpression die 9 months earlier than patients with low CA IX expression. Considering the dismal outcome of GBM patients (average overall outcome approximately 15 months), we would not like to underestimate the CA prognostic impact.
We have added the following information: “Lastly, Cetin and coworkers verified the finding of prognostic value of CA IX in glioblastomas and showed that the median overall survival was longer in patients with low levels of CA IX expression (18 months) compared to patients with CA IX overexpression (9 months). In multivariate analysis, significant prognosticators included CA IX overexpression, Karnofsky performance scale, incomplete temozolomide treatment, and gross-total resection. There was a trend towards a longer median progression free survival in patients with low CA IX expression (8 months vs. 3.5 months), but this difference did not reach statistical significance (p=0.054).” (Lines 214-220)
Line 261, it is customary to list the NCT number of clinical trials.
Our comment: The NCT numbers have been added (lines 395 and 406)
I don’t understand why the authors have not discussed the preclinical & clinical work on using already-marketed CAII-IX inhibitors in gliomas, references 1 through 8 below. As mentioned NCT 02770378 has already reported preliminary positive findings in recurrent glioblastoma. These findings [1-8 below] must be part of any discussion of CA function in gliomas.
Our comment: Based on the reviewer´s comment we have added several examples to sections “pre-clinical studies” and “clinical trials” where CA inhibitors have been tested. (lines 289-333, 341-344, 354-376, 382-394, 400-413)
In a didactic piece like this a figure [a schematic or “cartoon”] showing how CA regulated extra- and intra- cellular pH must be given.
Our comment: We have added a new Figure 1 and the corresponding legend (Lines 751-760)
The authors discuss their own work [Neuro Oncol. 2008 Apr; 10(2): 131–138.] but I didn´t understand that work then, nor do I on re-reading now. Fig. 3 in that 2008 work seems to show CA IHC staining having marginal but some influence on OS but what exactly was the glioma studied? What are diffusely infiltrating astrocytomas? Did they or did they not include grade III and grade IV tumors? Or...?
Our comment 1: At the end of the follow-up period the OS difference of patients with low vs strong staining was approximately 20% (patients having strong CA XII positivity vs no or weak CA XII), p=0.010, and CA XII was also an independent prognosticator.
Our comment 2: Diffusely infiltrating astrocytomas comprise of grade II diffuse astrocytomas, grade III anaplastic astrocytomas, and grade IV glioblastomas. During the time of that publication, WHO 2007 classification of brain tumors was used, and “diffusely infiltrating astrocytoma” was a standard definition separating those tumors from well circumscribed astrocytomas.
Our comment 3: Like discussed before, changing all tumor types to WHO 2016 based definitions is not feasible, since it is not possible to gather all needed molecular information from previous studies. E.g. IDH mutation status was reported to be an important feature in 2008 (Parsons et al Science 2008 Sep 26;321(5897):1807-12).
As mentioned above the authors’ discussion of CAII/CAIX inhibitors [starting line 228] in past work is grossly and intolerable inadequate.
Our comment: Based on the reviewer´s comment we have added several examples to the sections “pre-clinical studies” and “clinical trials” where different CA inhibitors, such as the classical drug, acetazolamide, anticonvulsants, topiramate and zonisamide, and a CA IX inhibitor, SLC-0111, have been tested. (lines 289-333, 341-344, 354-376, 382-394, 400-413)
The reviewer provided several references to be added. They are now included in the text as References 19, 57, 78, 79, 80, 86, 87, 88, 89, 91, 93
Round 2
Reviewer 3 Report
Can the authors make the diagram [“cartoon”] clearer ? the red blobs at the exchanger pumps are not clear. Also there, MCT is defined in legend but AE and NBC are not. CAinter in red is not explained. I assume this is CAII in cytosol but that must be made clearer.
Line 81 is odd. “The choroid plexus is one of the sites in CNS where CAs are expressed. “ since the authors just finished saying that CAs are ubiquitous in brain tissue and brain tumors. I believe line 82 is incorrect. Patients given acetazolamide to reduce ICP continue to make CSF, just less. So “halted” should be replaced with reduced.
Minor language oddities are present but these need not be corrected, as the authors’ intent remains clear throughout the ms. F. ex. “3.1 Screening of CA II, CA IX and CA XII in different brain tumors…” more natural would be “Screening for…”
Grade 4 gliomas [glioblastomas, GB] are almost always pos. for CAIX.
Line 135 y pVHL is presented but not previously defined. This abbreviation is not well enough known to be listed without definition but the protein, von Hippel-Lindau protein, is.
Fig.2A, I don’t see staining where some of the arrows point. Greater magnification or re-placing the arrows needed. The micrographs add greatly but show unacceptable sloppiness that must be corrected:
Line 145 the designation “C” is missing. In Fig 2. The letter follows the caption, line 143 “Weak endothelial CA II staining (arrows) in grade III anaplastic astrocytoma (A).” but in Fig.3. Line 149 we see “C. Faintly CA XII-positive multinucleated tumor cell in the middle. “
Currently ongoing clinical trials of CAII CAIX inhibition in cancer might be briefly mentioned:
NCT03467360 using acetazolamide in SCLC
NCT03450018 Studying SLC-0111 in pancreatic ca.
Line 250 that fact that Birner et al used “aberration” is no excuse for the authors repeating this mistake. Use the more accurate: deletions, overexpression, underexpression, mutated form, or ..? Line 400 …”and for celecoxib”.
Author Response
We thank reviewer 3 for a very precise evaluation of our manuscript.
Reviewer 3.
Can the authors make the diagram [“cartoon”] clearer ? the red blobs at the exchanger pumps are not clear. Also there, MCT is defined in legend but AE and NBC are not. CAinter in red is not explained. I assume this is CAII in cytosol but that must be made clearer.
Figure edited (ion channels style)
Added (text): (including Na+/HCO3 cotransporters (NBCs) and Cl−/HCO3 exchanger (AEs))
Added (text): intracellular CA II (CAintra)
Line 81 is odd. “The choroid plexus is one of the sites in CNS where CAs are expressed. “ since the authors just finished saying that CAs are ubiquitous in brain tissue and brain tumors. I believe line 82 is incorrect.
Added: abundantly
Patients given acetazolamide to reduce ICP continue to make CSF, just less. So “halted” should be replaced with reduced.
Corrected as suggested (reduced).
Minor language oddities are present but these need not be corrected, as the authors’ intent remains clear throughout the ms. F. ex. “3.1 Screening of CA II, CA IX and CA XII in different brain tumors…” more natural would be “Screening for…”
Corrected as suggested (Screening for).
Line 135 y pVHL is presented but not previously defined. This abbreviation is not well enough known to be listed without definition but the protein, von Hippel-Lindau protein, is.
Added: The most widespread CA IX and XII staining of all tumors was detected in hemangioblastoma samples affected by non- functioning von Hippel-Lindau protein.
Fig.2A, I don’t see staining where some of the arrows point. Greater magnification or re-placing the arrows needed.
Comment: figure 2A edited (arrows and magnification).
Added: Figure 2. A. Weak endothelial CA II staining (arrows) in small capillaries of grade III anaplastic astrocytoma.
The micrographs add greatly but show unacceptable sloppiness that must be corrected:
Line 145 the designation “C” is missing.
Corrected as suggested (C).
In Fig 2. The letter follows the caption, line 143 “Weak endothelial CA II staining (arrows) in grade III anaplastic astrocytoma (A).” but in Fig.3. Line 149 we see “C. Faintly CA XII-positive multinucleated tumor cell in the middle. “
Corrected as suggested, now the caption is first.
Currently ongoing clinical trials of CAII CAIX inhibition in cancer might be briefly mentioned:
NCT03467360 using acetazolamide in SCLC
NCT03450018 Studying SLC-0111 in pancreatic ca.
Added: In addition to brain tumors, there are ongoing clinical trials on the combination of acetazolamide, radiotherapy, chemotherapy with platinum and etoposide in localized small cell lung cancer (https://clinicaltrials.gov/ct2/show/NCT03467360), and on combination of SLC-0111 and gemcitabine in metastatic pancreatic ductal cancer (https://clinicaltrials.gov/ct2/show/NCT03450018).
Line 250 that fact that Birner et al used “aberration” is no excuse for the authors repeating this mistake. Use the more accurate: deletions, overexpression, underexpression, mutated form, or ..?
Corrected/added: deletion or imbalance status
Line 400 …”/and for celecoxib”.
Added: including a CA inhibitor, celecoxib,
